# Influence of Stabilizing and Encapsulating Polymers on Antioxidant Capacity, Stability, and Kinetic Release of Thyme Essential Oil Nanocapsules

**DOI:** 10.3390/foods9121884

**Published:** 2020-12-17

**Authors:** Ricardo M. González-Reza, Humberto Hernández-Sánchez, Maria L. Zambrano-Zaragoza, Gustavo F. Gutiérrez-López, Alicia Del-Real, David Quintanar-Guerrero, Benjamín Velasco-Bejarano

**Affiliations:** 1Departamento de Ingeniería Bioquímica, Instituto Politécnico Nacional, Escuela Nacional de Ciencias Biológicas, UP Adolfo López Mateos, Ciudad de México CP 07738, Mexico; gonzalez.reza@comunidad.unam.mx (R.M.G.-R.); hhernan1955@gmail.com (H.H.-S.); ggutierrezl@ipn.mx (G.F.G.-L.); 2Laboratorio de Procesos de Transformación y Tecnologías Emergentes de Alimentos, Nacional Autónoma de México, Facultad de Estudios Superiores Cuautitlán, Universidad Cuautitlán Izcalli, Estado de México, Cuautitlán Izcalli CP 54714, Mexico; 3Centro de Física Aplicada y Tecnología Avanzada, Departamento de Ingeniería Molecular de Materiales, Campus Juriquilla, Universidad Nacional Autónoma de México, Querétaro CP 76230, Mexico; adelreal@unam.mx; 4Laboratorio de Posgrado en Tecnología Farmacéutica, FES-Cuautitlán, Universidad Nacional Autónoma de México, Cuautitlán Izcalli, Estado de México, Cuautitlán Izcalli CP 54745, Mexico; quintana@unam.mx; 5Laboratorio L-122 Sección de Química Orgánica, Departamento de Ciencias Químicas, FES-Cuautitlán, Universidad Nacional Autónoma de México, Cuautitlán Izcalli, Estado de México, Cuautitlán Izcalli CP 54745, Mexico; qfbbenjamin.velascob@cuautitlan.unam.mx

**Keywords:** food nanotechnology, polymeric nanoparticles, natural actives, polyvinyl alcohol, pluronic, poly-ε-caprolactone, ethyl cellulose

## Abstract

The release kinetics, stability, and antioxidant capacity of thyme essential oil polymeric nanocapsules as a function of encapsulating (poly-ε-caprolactone and ethylcellulose) and stabilizing (polyvinyl alcohol and Pluronic^®^ F-127) polymers were established. Samples were evaluated in terms of particle size, zeta potential, release kinetics, calorimetry, infrared spectra, antioxidant capacity, and diffuse reflectance. The particle size obtained was below 500 nm in all cases, ensuring nanometric size. Zeta potential as a function of the stabilizing polymer. Encapsulation efficiency was higher in the samples that contained ethyl cellulose (around 70%), associated with its affinity for the molecules contained in the essential oil. Differential scanning calorimetry revealed a strong dependence on the encapsulating polymers as a function of the melting temperatures obtained. Infrared spectra confirmed that the polymeric nanocapsules had the typical bands of the aromatic groups of thyme essential oil. The antioxidant capacity evaluated is a function exclusively of the active content in the nucleolus of the nanocapsules. Nanoencapsulation was not a significant factor. Diffuse reflectance revealed high physical stability of the dispersions related directly to the particle size and zeta potential obtained (either by ionic or steric effect). These findings confirm favorable characteristics that allow proposing these systems for potential applications in food processing and preservation.

## 1. Introduction

Essential oils are of great commercial interest for their potential applications as fragrances in foods, cosmetics and use as active ingredients in insecticides and the pharmaceutical industry. Their use is sometimes challenging due to low water solubility, volatility, and stability. Micro- and nanoencapsulation of essential oils or their active ingredients are significant for applying essential oils in food and other industries. Thyme essential oil of *Thymus vulgaris* L. contains molecules of phenolic compounds, such as thymol and carvacrol, which give it its characteristic antioxidant and antimicrobial properties [1,2]. The antioxidant capacity of various systems has been studied, as well as the factors that directly influence its numerical value; among these is the composition (if it is an individual component or if it is mixed with other ingredients, whether antioxidants or not), the temperature at which the substance is found if it is in the presence of an atmosphere rich in oxygen or it is in a modified atmosphere, and, finally, the concentration of the antioxidant substance in the medium [3]. The potential application of this essential oil in food has been limited due to its low solubility in water and high volatility. Nanotechnology has allowed nanostructures of bioactive substances like essential oils to increase their solubility and allow targeted delivery, depending on their intended purpose [4]. One way in which these oils can be encapsulated is by forming polymeric nanocapsules that consist of an internal liquid core surrounded by a polymeric membrane [5]. Nanoparticles of this kind can be either lipophilic or hydrophobic, depending on the materials incorporated and the method used to obtain them [6]. For the formation of polymeric nanocapsules by the emulsification–diffusion process, the saturated organic phase, the saturated aqueous phase, and finally, the diluted phase is required [7]. According to the above, the organic phase saturated with water contains the bioactive compound (generally dissolved in an oily core) and the encapsulating polymer. The aqueous phase saturated with the organic phase contains the stabilizing polymer; finally, the dilution is carried out with the unsaturated aqueous phase, usually water. [8]. Polymers commonly used in active substances’ encapsulation include poly-ε-caprolactone (PCL) and ethyl cellulose (ETC). PCL is a biodegradable polyester approved by the U.S. Food and Drug Administration (FDA) that significantly impacts lipophilic active substances’ encapsulation due to its biocompatibility, slow degradation rate ease of processing [9]. Ethylcellulose is a biocompatible polymer derived from cellulose in which ethyl ether groups replace some of the hydroxyl groups. It is soluble in organic solvents but not in aqueous solutions, so it is highly-recommended for encapsulating lipophilic substances [10]. Polyvinyl alcohol (PVA) stands out among stabilizing polymers as a semi-crystalline polymer that is highly hydrophilic, biodegradable, biocompatible, and has good thermal properties, high water solubility, and good gas permeability, among other properties. PVA is produced by hydrolysis from different degrees of polyvinyl acetate. Its final polymer properties depend on the degree of hydrolysis [11]. Pluronic^®^F-127 (poloxamer 407) is a nonionic hydrophilic surfactant, a three-block copolymer consisting of a central hydrophobic block of polypropylene glycol flanked by two hydrophilic blocks of polyethylene glycol. Poloxamer is widely used as a stabilizer in nanostructured systems [12]. The choice of polymers for forming nanostructured systems depends directly on the application for which they are proposed. Maybe a feature most important of these systems is the controlled release of active substances. The modeling of the release profiles provides valuable information that can be used to predict storage times, food shelf life, and the formulation of intelligent and active packaging systems [13]. Thyme essential oil nanoencapsulation has been reported with other biopolymers such as chitosan, Arabic gum, casein, maltodextrin, and starch. However, the present proposal for a nanostructured system offers potential application in different branches of food processing, aiming to use biodegradable polymers widely used in the pharmaceutical industry, which is also safe (approved by the FDA) offers controlled release characteristics. These physically stable systems have already been tested in food applications such as edible coatings in fresh-cut fruits, using active substances such as β-carotene and α-tocopherol [14,15]. This investigation aimed to establish the physical characteristics, antioxidant capacity, and release profiles obtained for nanostructured thyme essential oil using different encapsulating and stabilizing polymers for potential applications in the food industry.

## 2. Materials and Methods

### 2.1. Materials

Thyme essential oil (ρ = 0.917 g/cm at 25 °C) was used to form the oil nucleus. Poly-ε-caprolactone (PCL) (MW ≈ 80 kDa, ρ = 1.147 g/cm^3^ at 25 °C) and ethylcellulose (ETC) (μ = 100 cP in 80:20 toluene/ethanol solution (50 g/L) at 25 °C, 48% ethoxy) were used as encapsulating polymers. Pluronic^®^ F-127 (Poloxamer 407) and polyvinyl alcohol (PVA) (M.W. ≈ 89–98 kDa, μ = 11.6-15.4 cP in 40 g/L H_2_O at 20 °C) were the stabilizing agents. ABTS (2,2’-azino-bis-(3-ethylbenzthiazoline)-6-sulfonic acid), DPPH (2,2-diphenyl-1-picrylhydracil) and FRAP (2,4,6-tri (2-pyridyl)-s-triazine) were used for antioxidant capacity measurement. The reagents were purchased from Sigma-Aldrich^®^ (St. Louis, MO, USA). High Performance Liquid Chromatography (HPLC)-grade organic solvents (ethyl acetate for the formation of the nanocapsules and cyclohexane for the release kinetics) used in the present experimental study were purchased by Fermont^®^, Monterrey, Mexico. The rest of reagents were analytical grade.

### 2.2. Nanocapsule Preparation

The emulsification–diffusion method was used to form the nanocapsules that contained thyme essential oil [7,8]. Ethyl acetate saturated with water was used as the organic phase, in which the encapsulating polymer (ethylcellulose or PCL) and thyme essential oil were dissolved. The aqueous phase consisted of water saturated with ethyl acetate, which contained the stabilizing polymer (polyvinyl alcohol or Pluronic^®^ F-127). The organic phase and the aqueous phase were subjected to ultra-high agitation in an ultraturrax (T18, IKA^®^ Werke, Staufen, Germany) at 420 s^−1^ for 10 min. Subsequently, the diffusion was promoted by adding excess water, following the ultra-high agitation to the aforementioned conditions. The excess organic solvent was removed on a rotary evaporator (RV10, IKA^®^ Wilmington, NC, USA) at 9 kPa and 45 °C. Once the samples were obtained, they were stored at 25 °C for later analysis. The nanocapsules obtained contained 1 g/L of thyme essential oil, 0.4 g/L of encapsulating polymer, and 4.8 g/L of stabilizing polymer.

### 2.3. Dynamic Light Scattering and Electrophoretic Movement

For the analysis of particle size (P.S.) and polydispersity index (PDI), the nanocapsules were diluted in distilled water (1:20) to obtain volume frequency histograms. For the evaluation of the dynamic light scattering, a fixed measurement angle of 273° was used. The study of the electrophoretic movement in the nanocapsules was expressed as zeta potential (ζ). For the evaluation of the ζ, the sample was diluted with distilled water of a similar form that the users in the dynamic light scattering measurement was carried out. The aforementioned parameters were analyzed on a Z-sizer 4 (Malvern Ltd., Grovewood Road, UK). Measurements of each parameter were carried out in independent experiments at 25 °C in triplicate [16].

### 2.4. Nanocapsule Morphology

The nanocapsules’ morphology containing thyme essential oil was analyzed using the methodology proposed by González-Reza et al. (2018) [4] by Scanning Electron Microscope (SEM). The images were observed in a scanning electron microscope (Hitachi, SU-8230, Tokyo, Japan) with a BSE + BSE (U) detector.

### 2.5. Encapsulation Efficiency (E.E.)

E.E. was determined by spectrophotometry using a Biospectrophotometer (Eppendorf AG, 22331, Hamburg, Germany) at 275 nm. To physically separate unencapsulated amounts of thyme essential oil was employment centrifugation in an ultracentrifuge (Hermle Z323K, Labortechnik GMBH, Wehingen, Germany) for 20 min at 4 °C and 18,000 ×*g*. E.E. was determined by the following expression:(1)$$EE\left(\%\right)=\left(\frac{Essential\;oil\;concentration\;retained}{Total\;concentration\;of\;essential\;oil}\right)\times 100$$

### 2.6. Release Kinetics

Release of thyme oil was carried out according to the methodology proposed by Soares et al. (2016) [17] with some modifications. Briefly, the release profiles of the N.C.s were determined using cyclohexane. The nanoparticles were lyophilized in a vacuum dryer. The resulting powder was resuspended in the cyclohexane according to treatment at a concentration of 1 g/L and separated by ultracentrifugation at 18,000× *g* for 20 min at 5 °C to remove suspended particles and avoid errors in the measurements. The volume lost periodically by each measurement was replaced. All samples’ absorbance was measured and correlated with thyme essential oil calibration curves (λ = 275 nm). The concentrations and the percentage of the active ingredient released over time were calculated in a Biospectrophotometer (Eppendorf AG, 22331, Hamburg, Germany). Controls were made by centrifuging 1 mL of free polymer in the release medium. All nanocapsule release experiments were performed in triplicate for 48 h at 25 °C.

### 2.7. Infrared Spectroscopy

The infrared absorption spectra of the individual compounds and the nanoparticles obtained were examined by spectroscopy in an I.R. spectrum (PerkinElmer Spectrum 400 IR, Waltham, MA, USA). Samples with a minimum moisture content were placed in the prism for further analysis. Spectra were obtained with an interval of 500–4000 cm^−1^ and a resolution of 1 cm^−1^ at 25 °C. The ambient spectrum was taken as a target and reference. The results obtained were analyzed in the equipment’s software

### 2.8. Thermal Behavior

The heat flow in the formed nanocapsules, as well as in their components, was obtained in a differential scanning calorimeter (T.A. Instruments, DSC Discovery, New Castle, DE, USA). The working conditions used were a measurement range of –20 to 150 °C with a heating ramp of 10 °C/min. The results obtained were analyzed in the equipment’s software.

### 2.9. Diffuse Reflectance

The stability of the nanocapsules was determined with a diffuse reflectance using Turbiscan MA2000 equipment (Formulaction, Toulouse, France), according to the measurement protocol proposed by González-Reza et al. (2018) [4] every 8 min for 24 h, utilizing two synchronous detectors: one for transmission and one for backscatter. Samples were analyzed at 25 °C [4].

### 2.10. Antioxidant Capacity

The antioxidant capacity of the nanocapsules was determined spectrophotometrically in a BioSpectrometer^®^ (Eppendorf AG, 22331, Hamburg, Germany) using positive control ascorbic acid by ABTS, DPPH, and FRAP. The results were contrasted with a reference curve constructed by plotting absorbance values against 100–1000 μmol concentrations of ascorbic acid. Results were expressed as μmol equivalents of ascorbic acid/g of nanostructured essential oil. Measurements were made in triplicate.

#### 2.10.1. ABTS

The antioxidant capacity determined by ABTS was evaluated according to the measurement protocol reported by Re et al. (1999) [18]. Measurements were made at 25 °C after 6 min by spectrophotometry at 734 nm.

#### 2.10.2. DPPH

The antioxidant capacity determined by DPPH was evaluated according to the measurement protocol reported by Brand-Williams et al. (1995) [19]. Measurements were made at 25 °C after a 30 min incubation time in the dark by spectrophotometry at 517 nm.

#### 2.10.3. FRAP

The antioxidant capacity determined by FRAP was evaluated according to the measurement protocol reported by Benzie & Strain (1996) [20]. Measurements were made at 37 °C after a 30 min incubation time in the dark by spectrophotometry at 595 nm. A control without a sample was also prepared.

### 2.11. Statistical Analyses

Statistically significant differences were established by ANOVA (α = 0.05). All statistical analyses were performed using the Minitab statistical program (Minitab^®^ Statistical Software 19 Inc., Center, PA, USA).

## 3. Results

### 3.1. Dynamic Light Scattering, Electrophoretic Movement, and Charging Efficiency

Table 1 shows the physical parameters obtained for the N.C.s under the different experimental conditions. All the samples were determined to have average particle sizes of 500 nm within nanometric size.

The particle size distribution obtained for the different nanostructured systems is shown in Figure 1. The samples presented significant differences (*p* ≤ 0.05) in the encapsulating polymer and stabilizing polymer used. Reviews about P.S. (less than 500 nm) of polymeric nanocapsules formed by the emulsification–diffusion method have been reported [6]. In addition to analyzing the nanoencapsulation of bioactive substances in polymers such as poly-ε-caprolactone and ethyl cellulose [4,21]. The polydispersity index values obtained—below 0.15 in all cases (*p* > 0.05)—suggest that the dispersions had a narrow distribution [22].

The ζ obtained for the samples stabilized with Pluronic^®^ F-127 was higher than that of the samples containing PVA (*p* ≤ 0.05), since in colloidal systems the charge arises from the ionization of the surface groups, adsorption of the material from the active surface, the continuous charges associated with the crystalline structures, or the combination of these mechanisms [23]. The lowest zeta potential values were found in the samples stabilized with PVA since that stabilization occurred as a result of steric effects rather than repulsion effects caused by surface charges [24]. In the case of encapsulation efficiency, values ranged from 60–70% for all samples, highlighting that the highest retention occurred using ethyl cellulose (*p* ≤ 0.05). The results were very near those reported by Galindo-Pérez et al. (2018) [21] to encapsulate essential oils using the emulsification–diffusion method.

### 3.2. Nanocapsule Morphology

Figure 2 shows micrographs that correspond to the polymeric nanocapsules, revealing a distorted, spherical shape with some agglomerations. The capsular structure consists of the polymeric membrane as a layer that surrounds an oily nucleus, as reported by Noronha et al. (2013) [25] for the formation of α-tocopherol/PCL nanocapsules by nanoprecipitation.

The average size of the nanostructure observed by SEM correlated successfully with laser light-scattering’s average size. The micrographs obtained reveal regular spherical nanocapsules, and their size agrees with that obtained by dynamic light scattering.

### 3.3. Release Kinetics of Nanocapsules

A wide variety of mathematical models are available to fit bioactive agent release data, most of them presented as nonlinear equations. Figure 3 shows the release profiles of the nanocapsules. The ANOVA revealed that the use of different encapsulating polymers gave a statistically significant effect (*p* ≤ 0.05) on the release of the thyme essential oil, in contrast to the stabilizing polymers (*p* > 0.05).

Regarding the modeling of the release profiles, five models widely used in the literature were tested to understand the phenomena that predominated in releasing the thyme essential oil in the nanocapsules formed by the diffusion emulsification method. Table 2 shows the kinetic parameters obtained from the thyme essential oil release curves.

The release kinetics modeled by Korsmeyer-Peppas presented a good correlation with all the models studied, as no statistical differences were found among the release constants (*p* > 0.05). Thyme essential oil release studies verified this behavior for chitosan nanocapsules prepared by nanoprecipitation [26]. The percentage of bioactive release can be determined using constant values. The “n” values in the Korsmeyer-Peppas model provides information on the release mechanism. If *n* < 0.43, there is Fickian diffusion [17]. Release studies conducted for polymeric nanocapsules (PCL/β-carotene) obtained by the emulsification–diffusion method and applied as an edible coating on fresh-cut melon found that the Korsmeyer-Peppas model indicated a release pattern that can be explained by time-dependent Fickian broadcast Zambrano-Zaragoza et al. (2017) [14]. This is consistent with the findings obtained in the present study.

### 3.4. Infrared Spectra of the Nanocapsules

Figure 4a shows the infrared (I.R.) characterization of the thyme essential oil. The typical thyme essential oil absorption peaks detected at 2926 and 2969 cm^-1^ were caused by the C-H stretching vibration of aliphatic methylene for the oil core. The peaks at 1586 and 1466 cm^−1^ were attributed to the C=C skeletal vibration of the oil’s benzene ring. The peaks detected at 1376 and 1108 cm^−1^ were assigned to the C-O-H flex mode and the -C-O- stretching vibration, respectively. The absorption peak at 947 cm^−1^ was attributed to the C=C stretching vibration or the bending vibration of some hydrogen-containing groups [27].

Figure 4i shows the spectrum of the PCL. Strong bands were as the carbonyl stretch mode around 1721 cm^−1^. The 1278 cm^−1^ band is assigned to the C-C and C-O spinal stretch modes in crystalline PCL [28]. The ETC (Figure 4h) presents a weak stretch band at 3480 cm^−1^ that corresponds to the O.H. bond vibration mode and medium intensity stretch bands at 2974 and 2871 cm^−1^ that, in turn, correspond to the C-bond vibration modes. C.H. is present in the alkane groups of the ethylcellulose. Typical stretch bands of 1444, 1375, and 1310 cm^−1^ were found that correspond to the vibratory mode of flexion of the -CH_3_ groups and corresponding 1444 cm^−1^ of the -CH_2_ groups, in addition to the characteristic band at 1052 cm^−1^ that corresponds to the vibration mode of the stretching of C.O. bonds [10]. The main peaks for Pluronic^®^ F-127 (Figure 4f) were present in the 2883 cm^−1^ (C.H. stretch), 1342 cm^−1^ (O.H. stretch), and 1099 cm^−1^ zones (C.O. stretch) and harmonized with the standard spectra of the aforementioned polymer. In the PVA case (Figure 4g), the bands at 3290 and 2911 cm^−1^ were attributed to O.H. and CH_2_ stretching, respectively. Other characteristic bands for PVA appeared at 1428, 1325, 1087, and 944 cm^−1^ and were attributed to O-H, C-H flexion, C-O-H flexion, and C-O and CH_2_ balancing, respectively.

The infrared spectra for polymeric nanocapsules were also analyzed. The spectra obtained at the typical absorption peaks of thyme essential oil were detected at 2926 and 2969 cm^−1^ in all samples. The peaks at 1563 and 1466 cm^−1^ were attributed to the C=C skeletal vibration of the benzene ring in the oil loaded in the nanocapsules. The peaks at 1374 and 1110 cm^−1^ were assigned to the C-O-H flex mode and C-O stretching vibration. The absorption peak at 947 cm^−1^ was attributed to the stretching vibration of C=C or the bending vibration of some hydrogen-containing groups. This confirms the presence of the characteristic functional groups of thyme essential oil in the samples. These displaced peaks indicate the intermolecular interaction between the polymers and the thyme essential oil. Intermolecular interaction plays an important role in the encapsulating polymer and the stabilizer because it is understood as evidencing compatibility in nanocapsules.

### 3.5. Thermal Analysis of the Nanocapsules

DSC analyzed the thermal behavior of the polymeric nanocapsules to confirm the formation of the thyme essential oil nanocapsules. Figure 5a–d shows the DSC thermograms of the stabilizing and encapsulating polymers used in this process. Figure 5a shows that the melting point of the PCL was 61 °C, similar to that reported by Wang et al. (2018) [9]. Ethylcellulose is a highly amorphous or even completely amorphous material. The Tg of ethylcellulose is in the range of 130–150 °C (Figure 5b), depending on the type and measurement method used [10]. In the case of the stabilizing polymers, the thermogram of Pluronic^®^ F-127 (Figure 5c) reveals an acute endothermic peak in the 50–70 °C range due to dehydration of this polymer. This surfactant does not show any endothermic peak or phase transition since it is an amorphous polymer and, therefore, exhibits a wide endotherm due to dehydration [12]. The PVA thermal curve (Figure 5d) presented two melting points at 55 °C and 144 °C, respectively. Studies carried out by Rebia et al. (2018) [11] indicate a similar behavior to that obtained in the present work.

Figure 5e–h shows the thermal curves obtained for the different nanocapsules obtained by the emulsification–diffusion method. They denote that the melting points are related directly to the encapsulating polymers, regardless of the stabilizing polymer. The decrease in peak amplitude and melting temperature values can be explained as a strong interaction between PCL, ETC that is independent of the essential oil, as described by Adeli (2016) [12] in trials with Pluronic^®^ and irbesartan. The quantitative characterization of the thermal parameters of the nanocapsules by DSC was also carried out. The ANOVA performed revealed no statistically significant difference for the T_g_ and T_Fusion_ of the samples (*p* > 0.05) with the same encapsulating polymer. However, in the case of ΔH, there was a significant difference (*p* ≤ 0.05) due only to the stabilizing polymer. PVA values are in the range of 2.5–3 kJ/kg and 75–81 kJ/kg for Pluronic^®^. The decrease in fusion enthalpy indicates a reduction in the crystallinity of the samples [11].

### 3.6. Stability of the Nanocapsules

Figure 6 presents the diffuse reflectance profiles obtained for the nanocapsules prepared by the emulsification–diffusion method. No statistically significant differences (*p* > 0.05) were detected between the samples in the different formation conditions (encapsulating and stabilizing polymers). Excellent physical stability has been reported for polymeric nanocapsules using poly-ε-caprolactone as encapsulating polymer, β-carotene as active substance (dissolved in sunflower oil), and Pluronic^®^ F-127 as stabilizing polymer during storage at 4 and 25 °C for 28 days [4]. This is consistent with the present experimental study. It is worth mentioning that in the present study, the good physical stability revealed in the diffuse reflectance studies is attributed to the difference in surface charges when Pluronic^®^ F-127 is used and to steric effects when polyvinyl alcohol is used.

### 3.7. Antioxidant Capacity of the Nanocapsules

The antioxidant capacity of the polymeric nanocapsules was established using different substances since each one provides important information on the possible action mechanisms of the contained substances (Table 3). This analysis revealed that, regardless of method, no significant difference (*p* > 0.05) was found between the antioxidant capacity obtained from the polymeric nanocapsules and that of the free essential oil. This implies that, in the sizes obtained (200–500 nm), there was no increase or decrease in this property and that the antioxidant capacity evaluated depends exclusively on the active molecules contained in the essential oil [1], regardless of the polymers used (whether encapsulating or stabilizing). The gas chromatography analysis performed by Oliveira et al. (2020) [29] indicated that the main active substances present in the essential oil of thyme were a total of 28 compounds, presenting five among the most abundant: thymol (50.5%), p-cymene (19.4%), γ-terpinene (9.1%), carvacrol (5.35%) and β-linalool (3.36%).

As reported in the literature, the antioxidant capacity may differ depending on the assay used. The antioxidant capacity evaluated by ABTS was greater in all cases than the results obtained by DPPH (less than 90%) and lower than 50% on the FRAP tests (*p* ≤ 0.05). This can be correlated to the ABTS assays that determine the hydrophilic and lipophilic antioxidant activity in food and provide a highly-adequate context for the overall antioxidant content [30]. However, the DPPH and FRAP test determinations in the present study allow us to understand various chemical phenomena involved in releasing the active substances from thyme essential oil, such as chelating capacity and polymerization properties.

Several studies have analyzed the antioxidant capacity of the major compounds in thyme essential oil. Milos and Dakota (2012) [31] carried out an investigation on antioxidant synergisms and antagonisms between thymol, thymoquinone, carvacrol, and p-cymene in a model system, finding that there is an antioxidant synergism between thymol and carvacrol and antagonism between thymoquinone and p-cymene. This allows inferring the variation between the different values evaluated for antioxidant capacity depending on the method (mechanism of action).

## 4. Conclusions

It was possible to obtain physically stable polymeric nanoparticles of nanometric size (less than 500 nm) with a low probability of aggregation in all cases. Thermal and I.R. analyses revealed good integration of the components in the nanostructured system, independent of the encapsulating and stabilizing polymers used. The study determined that the most adequate kinetic modeling of nanocapsule release for describing the phenomena involved is the one proposed by Korsmeyer-Peppas, which defines the release as Fickian in the nanostructured system. The nanocapsules’ antioxidant capacity reveals the different action mechanisms suggested for each test, such as polymerization and chelation. It was possible to form a nanostructured system with biodegradable wall polymers (PCL and PVA) that positively impacted the physical, antioxidant, and stability properties. The potential application of the nanostructured systems obtained in this work in food processing and preservation can be inferred.

## Figures and Tables

**Figure 1 foods-09-01884-f001:**
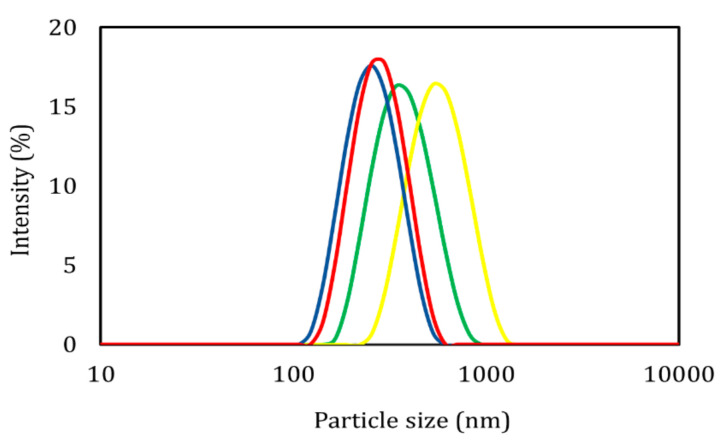
Distribution of the particle size obtained for the different treatments.

**Figure 2 foods-09-01884-f002:**
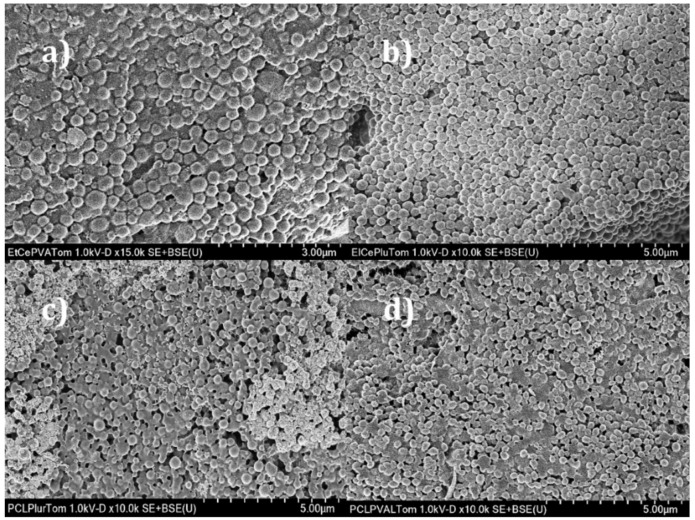
Morphological characterization by SEM of the polymeric nanocapsules: (**a**) ethyl cellulose-PVA; (**b**) ethyl cellulose-Pluronic^®^ F-127; (**c**) PCL-Pluronic^®^ F-127; (**d**) PCL-PVA.

**Figure 3 foods-09-01884-f003:**
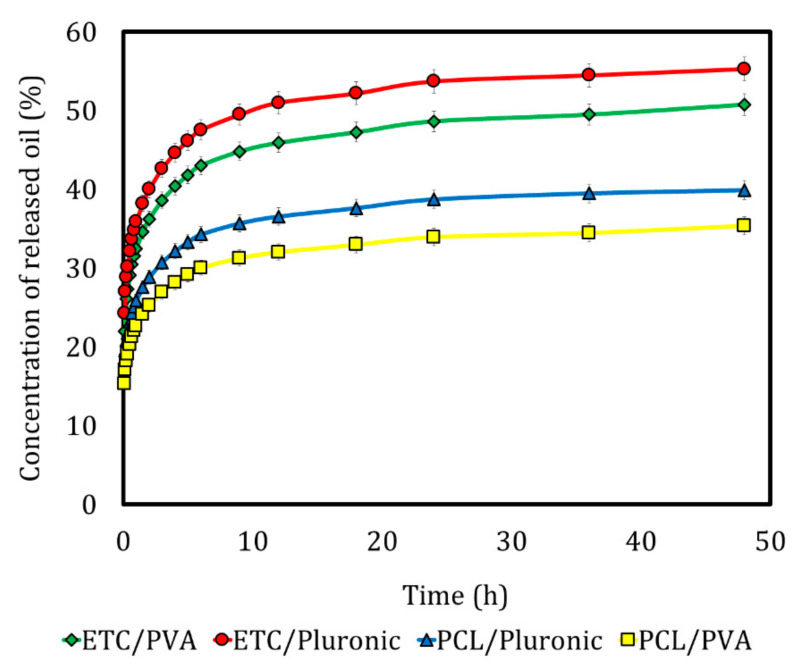
In vitro release profiles obtained for the different polymeric nanocapsules.

**Figure 4 foods-09-01884-f004:**
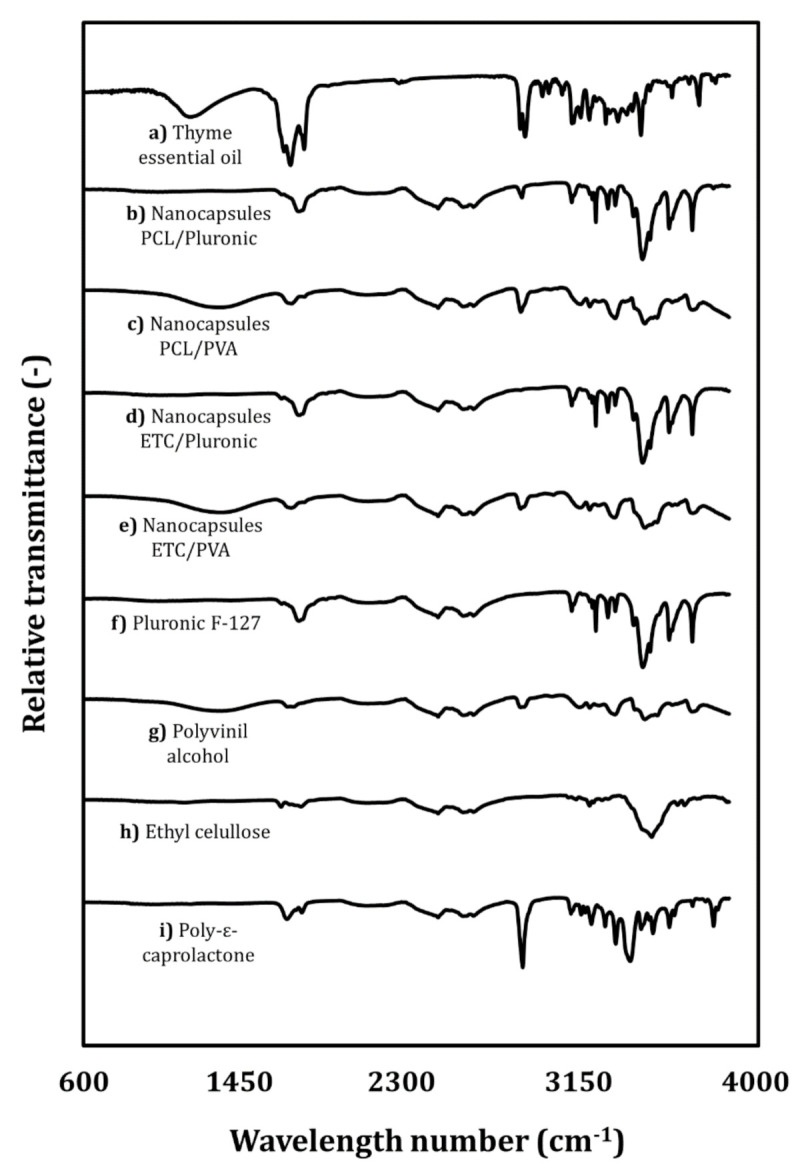
Infrared spectra for (**a**) thyme essential oil; (**b**) PCL-Pluronic^®^ F-127; (**c**) ETC-Pluronic^®^ F-127; (**d**) PCL-PVA) polymeric nanocapsules; the (**e**) ETC-PVA)and (**f**) Pluronic^®^ F-127 stabilizing polymers; and the (**g**) PVA; (**h**) ethyl cellulose; and (**i**) poly-ε-caprolactone encapsulating polymers.

**Figure 5 foods-09-01884-f005:**
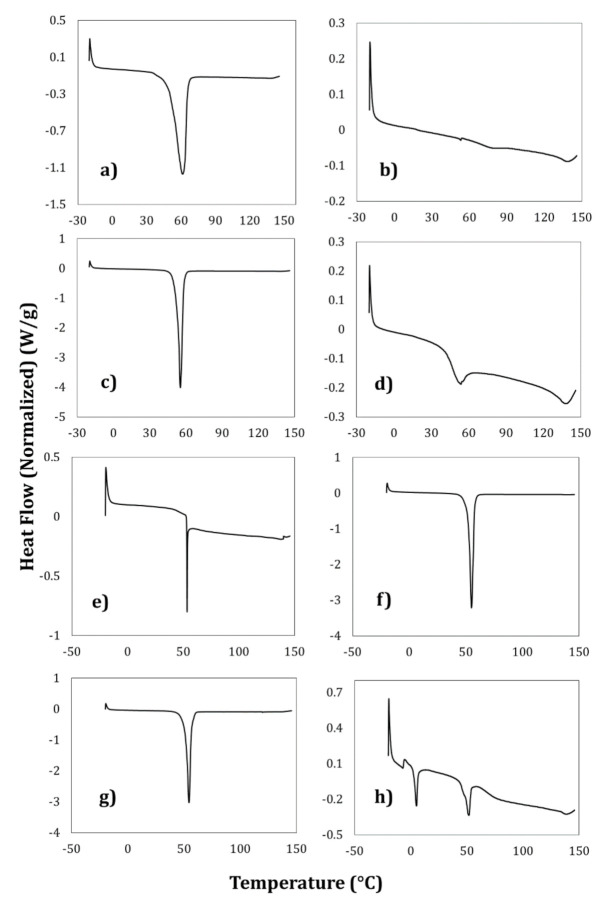
Differential scanning calorimetry for the encapsulating (**a**) poly-ε-caprolactone and (**b**) ethyl cellulose polymers, and the stabilizing polymers (**c**) Pluronic^®^ F-127 and (**d**) PVA used to form the polymeric nanocapsules: (**e**) ETC-PVA; (**f**) ETC-Pluronic^®^ F-127, (**g**) PCL-Pluronic^®^ F-127 and (**h**) PCL-PVA.

**Figure 6 foods-09-01884-f006:**
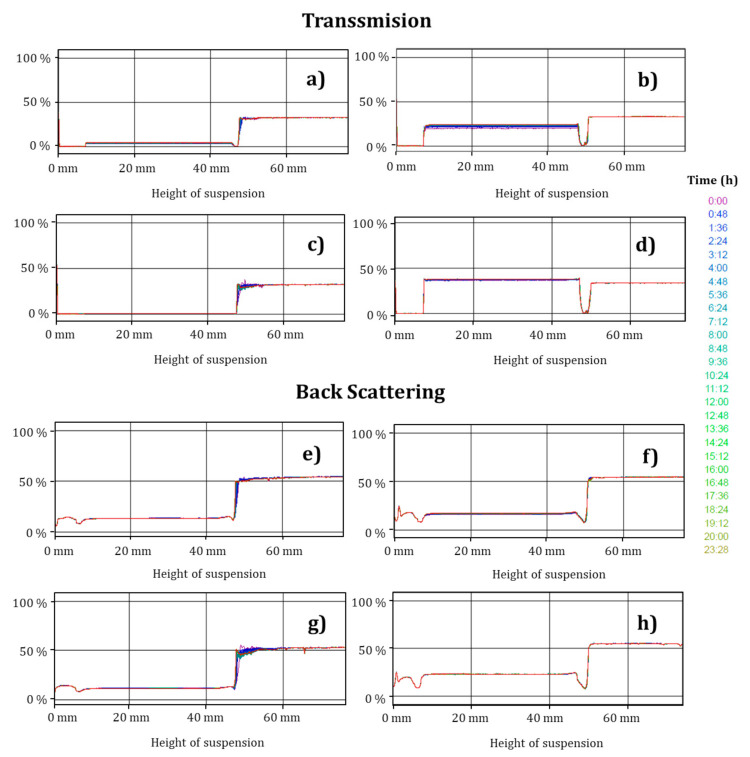
Diffuse reflectance: (**a**) and (**e**) ETC-PVA; (**b**) and (**f**) ETC-Pluronic^®^ F-127; (**c**) and (**g**) PCL-Pluronic^®^ F-127; and (**d**) and (**h**) PCL-PVA.

**Table 1 foods-09-01884-t001:** Characterization by laser light scattering, electrophoretic movement, and encapsulation efficiency of polymeric nanocapsules.

Sample	P.S. (nm)	PDI (-)	ζ (mV)	E.E. (%)
ETC-PVA	346.97 ± 0.85	0.09 ± 0.01	–7.16 ± 0.28	71.28 ± 0.68
ETC-Pluronic^®^	266.13 ± 3.53	0.11 ± 0.03	–19.77 ± 0.76	69.79 ± 2.92
PCL-PVA	489.20 ± 3.93	0.12 ± 0.01	–5.89 ± 0.36	60.87 ± 0.84
PCL-Pluronic^®^	241.37 ± 2.06	0.12 ± 0.05	–16.97 ± 0.83	67.43 ± 0.91

**Table 2 foods-09-01884-t002:** Models applied to the release profiles in polymeric nanocapsules.

Sample	Zero-Order	First-Order	Higuchi	Hixon- Crowel	Korsmeyer-Peppas	
	R^2^	R^2^	R^2^	R^2^	R^2^	n
ETC-PVA	0.5938	0.5116	0.8259	0.5395	0.9830	0.1363
ETC-Pluronic^®^	0.5797	0.5015	0.8259	0.5281	0.9808	0.1357
PCL-PVA	0.5925	0.5117	0.8259	0.5385	0.9830	0.1360
PCL-Pluronic^®^	0.5840	0.5049	0.8259	0.5316	0.9815	0.1357

**Table 3 foods-09-01884-t003:** Antioxidant capacity of the different polymeric nanocapsules obtained by the radicals ABTS, FRAP, and DPPH.

Sample	ABTS	FRAP	DPPH
	μmol_ascorbic acid equivalents/_g_nanostructured oil_		
Thyme oil	2010.74 ± 43.94	946.04 ± 12.87	274.74 ± 37.71
ETC-PVA	1924.03 ± 74.52	966.38 ± 22.18	272.79 ± 41.46
ETC-Pluronic^®^	2135.58 ± 33.72	1071.75 ± 19.59	285.13 ± 14.28
PCL-PVA	1973.69 ± 59.01	1108.92 ± 56.37	280.46 ± 20.02
PCL-Pluronic^®^	2034.40 ± 55.28	1133.06 ± 18.97	269.30 ± 12.07

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
