# Peer review of "Influence of Stabilizing and Encapsulating Polymers on Antioxidant Capacity, Stability, and Kinetic Release of Thyme Essential Oil Nanocapsules"

_foods, 2020, doi:10.3390/foods9121884_

Round 1

Reviewer 1 Report

The research manuscript entitled “Influence of stabilizing and encapsulating polymers on antioxidant capacity, stability and kinetic release of thyme essential oil nanocapsules’’ by Gonzalez-Reza et al. is a very interesting manuscript in the field of food processing and preservation, in addition to dealing with a current and novel topic such as the encapsulation of essential oils. The methodology proposed for the NCs preparation as well as the characterization is adequate and allows for sufficient information to be obtained to draw conclusions from the work. I sincerely congratulate the authors for this very complete and well explained work. I think that to further enrich this manuscript some things could be added.

  • Please, check the manuscript and pay attention to some spelling errors. For example, in line 69: ‘highly hydrophilic’.
  • In section 2. Nanocapsule preparation, authors reference the method used but it could be helpful to briefly add the steps followed to prepare them. Specially, being a mixture of two protocols (Quintanar-Guerrero et al. 1998, and Zambrano-Zaragoza et al. 2011).
  • It seems that in sections 4. Infrared spectra of the nanocapsules and 3.5. Thermal analysis of the nanocapsules, the reference to the Figures 4 and 5 is wrong, due to the FTIR spectra is the Figure 4 and in section 3.4. is referring to Figure 5, instead of Figure 4. The same happens in section 3.5.
  • In section 6. Antioxidant capacity of the nanocapsules, as mentioned by the authors, the use of different methods to determine this property provide important information on the action mechanisms of the contained substances. However, it seems to me that more discussion of the results obtained is needed. Thyme essential oil has been widely characterized in literature so, no further tests to study the chemical composition of this essential oil is needed, but authors could correlate the antioxidant activities obtained through the different methods with the thyme essential oil chemical composition. Moreover, considering the data obtained and the relationship between antioxidant power and antimicrobial activity, authors could mention if the NCs are expected to exhibit biocide power, since it is an important parameter in the field of food preservation.  
  • Figure 6. Please avoid using Spanish words.
  • I think it would be appropriate to enlarge the Conclusions

Author Response

Reviewer 1.

  1. The research manuscript entitled "Influence of stabilizing and encapsulating polymers on antioxidant capacity, stability and kinetic release of thyme essential oil nanocapsules" by Gonzalez-Reza et al. is very interesting in the field of food processing and preservation, in addition to dealing with a current and novel topic such as the encapsulation of essential oils. The methodology proposed for the NCs preparation as well as the characterization is adequate and allows for sufficient information to be obtained to draw conclusions from the work. I sincerely congratulate the authors for this very complete and well-explained work. I think that to further enrich this manuscript some things could be added.

We appreciate the time and commitment of the reviewer to give us their comments and enrich the research article

  1. Please, check the manuscript and pay attention to some spelling errors. For example, in line 69: 'highly hydrophilic'.

The grammar error has been corrected, and it has been sent to the English language style review again to avoid future problems.

  1. In section Nanocapsule preparation, authors reference the method used but it could be helpful to briefly add the steps followed to prepare them. Specially, being a mixture of two protocols (Quintanar-Guerrero et al. 1998, and Zambrano-Zaragoza et al. 2011).

The methodology to be followed for nanocapsules' formation has been described, detailing what was requested by the reviewer.

  1. It seems that in sections 4—infrared spectra of the nanocapsules and 5. The nanocapsules' thermal analysis, the reference to Figures 4 and 5 is wrong due to the FTIR spectra in Figure 4 and in section 3.4. is referring to Figure 5, instead of Figure 4. The same happens in section 3.5.

The numbering of the figures has been corrected, adapting them to the text's sequence and the sections. We appreciate your review.

  1. In section Antioxidant capacity of the nanocapsules, as mentioned by the authors, the use of different methods to determine this property provide important information on the action mechanisms of the contained substances. However, it seems to me that more discussion of the results obtained is needed. Thyme essential oil has been widely characterized in literature so, no further tests to study the chemical composition of this essential oil is needed, but authors could correlate the antioxidant activities obtained through the different methods with the thyme essential oil chemical composition. Moreover, considering the data obtained and the relationship between antioxidant power and antimicrobial activity, authors could mention if the NCs are expected to exhibit biocide power, since it is an important parameter in the field of food preservation.

Added the chemical composition of thyme essential oil obtained by gas chromatography. Also, information has been correlated and added about how these compounds are directly responsible for the samples' antioxidant capacity. Although the antioxidant capacity of thyme essential oil is well known, it has not been reported in nanostructured systems such as those obtained in the present experimental study.

  1. Figure 6. Please avoid using Spanish words.

The figure has been corrected, and the document has been revised in its entirety to avoid incorrect words. We appreciate your review.

  1. I think it would be appropriate to enlarge the Conclusions

The conclusions have been expanded based on the results obtained in the present experimental research work

Reviewer 2 Report

Essential oils are of great commercial interest for their potential applications as fragances in foods, cosmetics, and use as active ingredients in insecticides and in the pharmaceutical industry. Their use is sometimes challenging due to poor water solubility, volatility, and stability. Micro and nanoencapsulation of essential oils or their active ingredients is of significant importance for the application of essential oils in food and other industries.

The authors have previously reported the preparation and characterization of nanocapsules for encapsulation of food ingredients (e.g., soybean, sunflower, grape oils) – Reference: . Zambrano-Zaragoza, M.L.; Mercado-Silva, E.; Gutiérrez-Cortez, E.; Castaño-Tostado, E.; Quintanar-Guerrero, D. Optimization of nanocapsules preparation by the emulsion-diffusion method for food applications. LWT - Food Sci. Technol. 2011, 44, 1362–1368, doi:10.1016/j.lwt.2010.10.004.  

 In this new manuscript, the authors have focused their efforts on the encapsulation of thyme essential oil in polymeric nanocapsules using polycaprolactone and ethyl cellulose for encapsulation and polyvinyl alcohol or Pluronic F-217 as stabilizing polymers.  Several characterization techniques were employed to assess the encapsulation efficiency, chemical characterization of encapsulated thyme oil, kinetics of release and antioxidant properties of the encapsulated material.   

However, the paper does not present sufficient data on how these new thyme essential oil nanocapsules could be applicable in the food industry.  A significant weakness is the evaluation of release kinetics in a solvent such as cyclohexane, an unlikely solvent to use for evaluation of potential application of nanocapsules for food applications.   

The following are specfic comments/suggestions requiring attention:

  1. What is the composition of the thyme essential oil employed in this work? The authors indicate in Lines 47-48 that “thyme essential oil contains molecules of phenolic compounds, such as thymol and carvacrol”.  Why do the authors highlight these two components? Thymol is the major component of thyme essential oil (46% thymol, but carvacrol is present at a low concentration (2.4%)  Reference:  https://earthwiseagriculture.net/wp-content/uploads/2020/06/Essential-Oil-Composition.pdf).  Several other terpenoid compounds in thyme essential oil are present in higher content than carvacrol and are likely to have an effect on the antimicrobial and antioxidant properties of thyme oil. 
  2. Introduction, Line 69: there is a typo, delete m from “highlymhydrophilic”
  3. Line 120: Sentence starting with “For to physically separate…” needs revision. It is confusing and contains few grammar/spelling errors.
  4. Line 123: Release kinetics experiments.  Why were the release profiles of the NCs determined in cyclohexane? This does not seem a suitable solvent to assess the possible benefits of NCs for food applications. Are the authors interested in assessing the stability of NCs during food storage? Or the controlled release of thyme oil upon digestion of food containing NCs?  Proper aqueous-based systems should be used to assess the potential benefits of these nanocapsules in foods. For example, if the authors are interested in assessing the controlled release of thyme oil then the NCs should be evaluated for stability of simulated gastric fluid and release in simulated intestinal fluid.
  5. Line 126. “The nanoparticles were lyophilized in a vacuum dryer”.  Nano and microparticles are prone to aggregation following lyophlization.  Is this an issue with the NC of thyme oil? Were particle size measurements obtained before and after lyophlization to determine potential particle aggregation?
  6. Lines 160, 167, 173. The subsections starting in these lines are subsections of 2.10 – Antioxidant activity. These subsections are incorrectly labeled 2.11.1, 2.11.2 and 2.11.3
  7. Incorrect figure numbers
    1. Figure 3 is incorrectly labeled as Figure 1 in the figure caption (Page 7)
    2. The infrared spectra is referenced in the text as Figures 5a-5h, but the IR spectra is in Figure 4
    3. The thermal analysis results are cited in the text as Figure 4a-4d. The data is shown in Figure 5.
  8. The nanoencapsulation of thyme essential oil has been reported in the literature using other biocompatible polymeric materials such as chitosan, Arabic gum, casein, maltodextrin, starch. What are the potential advantages of this new NCs compared to some of the other polymers previously employed for nanoencapsulation of thyme oil? 

Author Response

Reviewer 2.

  1. Essential oils are of great commercial interest for their potential applications as fragrances in foods, cosmetics, and use as active ingredients in insecticides and in the pharmaceutical industry. Their use is sometimes challenging due to poor water solubility, volatility, and stability. Micro and nanoencapsulation of essential oils or their active ingredients is of significant importance for the application of essential oils in food and other industries. The authors have previously reported the preparation and characterization of nanocapsules for encapsulation of food ingredients (e.g., soybean, sunflower, grape oils) – Reference: . Zambrano-Zaragoza, M.L.; Mercado-Silva, E.; Gutiérrez-Cortez, E.; Castaño-Tostado, E.; Quintanar-Guerrero, D. Optimization of nanocapsules preparation by the emulsion-diffusion method for food applications. LWT - Food Sci. Technol. 2011, 44, 1362–1368, doi:10.1016/j.lwt.2010.10.004.

We thank the reviewer for carrying out a retrospective analysis of the work carried out in our laboratory to comment on the manuscript.

  1. In this new manuscript, the authors have focused their efforts on the encapsulation of thyme essential oil in polymeric nanocapsules using poly-ε-caprolactone and ethyl cellulose for encapsulation and polyvinyl alcohol or Pluronic F-217 as stabilizing polymers.  Several characterization techniques were employed to assess the encapsulation efficiency, chemical characterization of encapsulated thyme oil, kinetics of release and antioxidant properties of the encapsulated material. However, the paper does not present sufficient data on how these new thyme essential oil nanocapsules could be applicable in the food industry.  A significant weakness is the evaluation of release kinetics in a solvent such as cyclohexane, an unlikely solvent to use for evaluation of potential application of nanocapsules for food applications. The following are specfic comments/suggestions requiring attention:

We regret that the manuscript is not suitable for reviewer number 2. However, we will make the adjustments as required by the reviewer; likewise, we will give strong arguments to justify the results. We are sure that with this it will be possible to publish our study in this important journal.

  1. What is the composition of the thyme essential oil employed in this work? The authors indicate in Lines 47-48 that "thyme essential oil contains molecules of phenolic compounds, such as thymol and carvacrol".  Why do the authors highlight these two components? Thymol is the major component of thyme essential oil (46% thymol, but carvacrol is present at a low concentration (2.4%)  Reference:  https://earthwiseagriculture.net/wp-content/uploads/2020/06/Essential-Oil-Composition.pdf).  Several other terpenoid compounds in thyme essential oil are present in higher content than carvacrol and are likely to have an effect on the antimicrobial and antioxidant properties of thyme oil. 

A reference to the chemical composition of thyme essential oil was added, which also addresses its relationship with the antioxidant capacity obtained, in the same way as requested by reviewer 1. We appreciate your observation. The additional references for this point are:

Oliveira, R.C.; Carvajal-Moreno, M.; Correa, B.; Rojo-Callejas, F. Cellular, physiological and molecular approaches to investigate the antifungal and anti-aflatoxigenic effects of thyme essential oil on Aspergillus flavus. Food Chem. 2020, 315, doi:10.1016/j.foodchem.2019.126096.

Milos, M.; Makota, D. Investigation of antioxidant synergisms and antagonisms among thymol, carvacrol, thymoquinone and p-cymene in a model system using the Briggs-Rauscher oscillating reaction. Food Chem. 2012, 131, 296–299, doi:10.1016/j.foodchem.2011.08.042.

  1. Introduction, Line 69: there is a typo, delete m from "highlymhydrophilic"

The grammar error has been corrected, and it has been sent to the English language style review again to avoid future problems.

  1. Line 120: Sentence starting with "For to physically separate…" needs revision. It is confusing and contains few grammar/spelling errors.

The grammar error has been corrected, and it has been sent to the English language style review again to avoid future problems.

  1. Line 123: Release kinetics experiments.  Why were the release profiles of the NCs determined in cyclohexane? This does not seem a suitable solvent to assess the possible benefits of NCs for food applications.

To establish the release kinetics of an active substance, it is necessary that the organic solvent it will use dissolves the bioactive, but not the encapsulating polymer, hence the importance of the release medium's choice. Our work team has published food application release studies using cyclohexane.

Zambrano-Zaragoza, M. L., Quintanar-Guerrero, D., Del Real, A., Piñon-Segundo, E., & Zambrano-Zaragoza, J. F. (2017). The release kinetics of β-carotene nanocapsules/xanthan gum coating and quality changes in fresh-cut melon (cantaloupe). Carbohydrate polymers, 157, 1874-1882.

  1. Are the authors interested in assessing the stability of NCs during food storage?

Regarding the storage stability of polymeric nanocapsules obtained by the emulsification-diffusion method, our work team has established physical stability for poly-ε-caprolactone as a wall polymer at different temperatures, different pH values, and β-carotene concentrations.

González-Reza, R. M., Quintanar-Guerrero, D., Del Real-López, A., Piñon-Segundo, E., & Zambrano-Zaragoza, M. L. (2018). Effect of sucrose concentration and pH onto the physical stability of β-carotene nanocapsules. LWT, 90, 354-361.

Therefore, the study of said stability could be subject to the fact that it has already been studied, and consequently, the work team decided not to include that part in the present manuscript.

  1. Or the controlled release of thyme oil upon digestion of food containing NCs?  Proper aqueous-based systems should be used to assess the potential benefits of these nanocapsules in foods.

One of the reasons we decided to evaluate the release in organic solvents with the aforementioned characteristics was that these results might be useful in the proposal and selection of nanostructured systems as active nano-packaging components biodegradable packaging and that also have an impact on increasing the shelf life of food.

  1. For example, if the authors are interested in assessing the controlled release of thyme oil then the NCs should be evaluated for stability of simulated gastric fluid and release in simulated intestinal fluid.

Digestibility studies are of great importance; however, calorimetry, physical stability, and antioxidant capacity tests lead the reader to believe that these systems (nanocapsules containing thyme essential oil) have useful properties for use in active packaging and increase shelf life of food. Although digestibility tests are adequate for evaluating its release in the tract, its nutritional impact was not addressed in the present study. The team that collaborated on the manuscript does not consider its integration in the paper adequate.

  1. Line 126. "The nanoparticles were lyophilized in a vacuum dryer".  Nano and microparticles are prone to aggregation following lyophlization.  Is this an issue with the NC of thyme oil? Were particle size measurements obtained before and after lyophlization to determine potential particle aggregation?

In order to avoid instability phenomena due to water in the samples, which was a determining factor in evaluating the release kinetics, it was decided to freeze the samples. SEM analyzed the samples to establish a change in their morphology or diameter; however, no statistically significant differences were found that was important to mention in the manuscript.

  1. Lines 160, 167, 173. The subsections starting in these lines are subsections of 2.10 – Antioxidant activity. These subsections are incorrectly labeled 2.11.1, 2.11.2 and 2.11.3

The numbering has been corrected; we thank the reviewer for the observation

  1. Incorrect figure numbers

The numbering of the figures throughout the manuscript has been corrected according to the text of the manuscript.

  1. Figure 3 is incorrectly labeled as Figure 1 in the figure caption (Page 7)

The figure numbering has been corrected.

  1. The infrared spectra is referenced in the text as Figures 5a-5h, but the IR spectra is in Figure 4

The numbering of the figures throughout the manuscript has been corrected according to the text of the manuscript.

  1. The thermal analysis results are cited in the text as Figure 4a-4d. The data is shown in Figure 5.

The numbering of the figures throughout the manuscript has been corrected according to the text of the manuscript.

  1. The nanoencapsulation of thyme essential oil has been reported in the literature using other biocompatible polymeric materials such as chitosan, Arabic gum, casein, maltodextrin, starch. What are the potential advantages of this new NCs compared to some of the other polymers previously employed for nanoencapsulation of thyme oil?

The importance of the polymers used in the present experimental study has been highlighted in a section highlighted in the introductory part.

Round 2

Reviewer 2 Report

The authors have incorporated statements in the introduction, results and discussion, and conclusion sections, and have added citations to back these new statements. These edits addressed my previous comments and provide good support to clarify the scope of their work related to development of nanoencapsulated oils for food storage stability.